# Deep Learning-Based Four-Region Lung Segmentation in Chest Radiography for COVID-19 Diagnosis

**DOI:** 10.3390/diagnostics12010101

**Published:** 2022-01-03

**Authors:** Young-Gon Kim, Kyungsang Kim, Dufan Wu, Hui Ren, Won Young Tak, Soo Young Park, Yu Rim Lee, Min Kyu Kang, Jung Gil Park, Byung Seok Kim, Woo Jin Chung, Mannudeep K. Kalra, Quanzheng Li

**Affiliations:** 1Transdisciplinary Department of Medicine & Advanced Technology, Seoul National University Hospital, Seoul 03080, Korea; younggon2.kim@gmail.com; 2Department of Radiology, Massachusetts General Hospital, Boston, MA 02114, USA; KKIM24@partners.org (K.K.); dwu6@mgh.harvard.edu (D.W.); HREN2@mgh.harvard.edu (H.R.); MKALRA@mgh.harvard.edu (M.K.K.); 3Department of Internal Medicine, School of Medicine, Kyungpook National University, Daegu 41944, Korea; wytak@knu.ac.kr (W.Y.T.); psyoung0419@gmail.com (S.Y.P.); deblue00@naver.com (Y.R.L.); 4Department of Internal Medicine, Yeungnam University College of Medicine, Daegu 42415, Korea; kmggood111@naver.com (M.K.K.); gsnrs@naver.com (J.G.P.); 5Department of Internal Medicine, Catholic University of Daegu School of Medicine, Daegu 42472, Korea; kbs9225@cu.ac.kr; 6Department of Internal Medicine, Keimyung University School of Medicine, Daegu 42601, Korea; chung50@dsmc.or.kr

**Keywords:** COVID-19, deep learning, segmentation, detection, lung, left hilum, carina, RALE

## Abstract

Imaging plays an important role in assessing the severity of COVID-19 pneumonia. Recent COVID-19 research indicates that the disease progress propagates from the bottom of the lungs to the top. However, chest radiography (CXR) cannot directly provide a quantitative metric of radiographic opacities, and existing AI-assisted CXR analysis methods do not quantify the regional severity. In this paper, to assist the regional analysis, we developed a fully automated framework using deep learning-based four-region segmentation and detection models to assist the quantification of COVID-19 pneumonia. Specifically, a segmentation model is first applied to separate left and right lungs, and then a detection network of the carina and left hilum is used to separate upper and lower lungs. To improve the segmentation performance, an ensemble strategy with five models is exploited. We evaluated the clinical relevance of the proposed method compared with the radiographic assessment of the quality of lung edema (RALE) annotated by physicians. Mean intensities of segmented four regions indicate a positive correlation to the regional extent and density scores of pulmonary opacities based on the RALE. Therefore, the proposed method can accurately assist the quantification of regional pulmonary opacities of COVID-19 pneumonia patients.

## 1. Introduction

The COVID-19 is a novel infectious disease, caused by severe acute respiratory syndrome coronavirus 2 (SARS-CoV-2), which could lead to acute respiratory distress syndrome (ARDS) [1,2]. Starting in December 2019, COVID-19 became a pandemic that has claimed over 811 thousand lives, infected over 47 million people worldwide, and wrecked economic and social hardships in all six inhabited continents [3]. Real-time reverse transcription-polymerase chain reaction (RT-PCR) is the preferred test for confirming COVID-19 infection. Currently, most international and national organizations recommend RT-PCR assays for screening and initial diagnosis of COVID-19 infection.

For the screening of severe COVID-19 pneumonia patients, computed tomography (CT) and chest radiography (CXR) are commonly used in sites with high prevalence. There is consensus that imaging should be used judiciously, and most often, in patients with moderate to severe disease and those with complications and comorbidities. Both CT and CXR are used for establishing the disease extent or severity of pulmonary opacities. Compared to CT, CXR is more accessible, mobile, cheaper, lower dose, efficient, and convenient to be utilized in intensive care settings. Prior studies have shown the crucial roles of imaging modalities for the initial diagnosis and quantification of the severity of COVID-19 pneumonia [2,4,5].

Deep learning-based methods with both CT and CXR have been explosively investigated in the automated classification and detection of COVID-19 [6,7,8,9]. For the robustness, an ensemble method is widely used [10]. To assess the disease severity from the quantitative extent of pneumonia, an automatic method for prediction of severity score have been introduced with a deep learning-based method [11], which showed high correlation scores at R [2] 0.865 and 0.746 with radiological extent and opacity, respectively. Clinical studies using CXRs have conducted to segment the whole lung into segments for predicting the severity of the disease [12]. With the importance of early diagnose of severity in CXRs, lung segmentation methods have focused on reducing non-specific signals such as tube or lines efficiently [13,14].

The studies using CXRs of COVID-19 patients found that the pneumonia ranges from normal lungs and subtle haziness in mild or early to more extensive diffuse opacities consistent with diffuse pneumonia and adult respiratory distress syndrome (ARDS) in severe and advanced disease. Radiographic assessment of lung edema (RALE) is a score indicating the severity of ARDS and COVID-19 pneumonia on CXRs [15]. For the RALE score, each lung is typically divided into upper and lower regions based on a horizontal line through the level of origin of the left upper lobar bronchus from the left mainstem bronchus. Then, the density and extent of pulmonary opacity are subjectively graded by radiologists in each of the four regions to determine the regional and total scores of pulmonary opacities. Particularly, the inclusion of lower lobes was reported in the majority of COVID-19 patients [16]. The RALE score has been validated as a good predictor of ARDS [12]. However, the method is prone to inter- and intra-observer variations, and inefficient for incorporating into interpretation routine. Other studies have shown a six-region division of lungs segmented manually for the diagnosis [17,18], which is also inefficient and requires a lot of effort. Therefore, an accurate sub-region segmentation method of the whole lung will be useful in clinics.

In this paper, to assist the diagnosis of lung severity with sub-regions, we propose a fully automated segmentation method using deep learning models to segment four regions of the whole lung in CXR. Specifically, two deep learning-based segmentation and detection models are combined as shown in Figure 1. To compute four-region lung masks, left and right regions are divided by the segmentation model, where a majority voting ensemble method is used from five different deep learning-based models to achieve a robust four-region lung segmentation in CXR. Then, the upper and lower sub-regions are divided by the positions of the carina and left hilum predicted by a deep learning-based detection model. To clinically evaluate the performance of the segmented regions, we compared the RALE scores of four regions done by physicians as labels with the mean intensity after normalization for each region, where the correlations with the extent and density scores of pulmonary opacities will be measured. In this paper, we focus on the accurate and robust four-region segmentation, and the robust RALE score estimation combined with our segmentation method will be developed in the near future.

## 2. Method

### 2.1. Segmentation Model

U-net architecture [19] using the skip connection was used to train the segmentation model, which is the most widely used network for segmentation in medical imaging. We trained five segmentation models with different conditions including backbones, pre-processing, and augmentation properties as shown in Table 1. EfficientNet v0 and v7 architectures [20] were used as the backbone networks in the U-net to train five (three v0 and two v7) segmentation models. Augmentations of Gaussian noise and gamma correction were applied to improve the robustness of the models to pixel noises from multiple CXR modalities. To train segmentation models for anterior-posterior (AP) CXRs that are not included in the public datasets, morphological transformation methods such as grid distortion, affine transform, and elastic transformation with different parameters were used in the augmentation process [21]. Five binary masks were used to generate the one ensemble mask based on the majority voting method. Technically, if half of the masks were predicted as a lung region, the pixel is labeled as the lung. The augmentation methods, such as affine transform, Gaussian noise, scaling, cutout, contrast, and brightness, were applied while training [5] of the five models. All models were trained with the same hyper-parameters, such as Adam optimizer (learning rate: 0.0001), epochs (200), batch size (8), and same input size at 256 × 256. Best models were selected at the lowest loss on the validation dataset. In addition, post-processing step was employed to refine the ensemble mask, where all the isolated holes were filled with the dilation operation.

### 2.2. Detection Model

We adopted a detection model to find a central point of the whole lung into four-regions such as right upper region (RUR), right lower region (RLR), low upper region (LUR), and left lower region (LLR). Although conventional RALE score described a horizontal line through the origin of the left upper lobe bronchus for the four-segments of lungs, it is difficult to identify the horizontal line in most of COVID-19 CXRs. Instead, the left hilum is the closest landmark for dividing upper and lower regions. However, the left hilum is sometimes difficult to be detected in patients with diseases progressed or patient rotation. In this case, the carina position can help to define the center position, which can be assumed at approximately 2 cm [22] above from the left hilum vertically. Therefore, we use the positions of left hilum and carina to accurately identify the central point for the horizontal lung segmentation into upper and lower regions. 

RetinaNet [23] was used for the detection model to detect positions of the carina and the left hilum. The central point of prediction box is used as a reference horizontal level that divide the upper and lower lungs. Specifically, we typically use the prediction box of the left hilum to divide the lung into upper and lower regions. However, if the model confidence of the left hilum detection is lower than 0.9, the position of 2 cm above from the prediction box of the carina is used.

To train the robust detection model, augmentation methods [23] such as rotation, translation, shearing, scaling, pixel noise, different range of contrast, brightness, hue, and saturation were used. The best model was selected as the lowest total loss in the validation set as shown in Table 2. The model performance was validated in the testing set.

### 2.3. Normalization

Intensity normalization is widely used as a pre-processing to make a similar statistical distribution of CXRs. Various scanners and setting parameters from multi-sites can cause a significant disparity of contrast and brightness of CXRs as shown in Figure 2. Density scores of Figure 2a–d were confirmed as zero by physicians while the disparity of mean intensities of the lung was large. To reduce this variation, intensity normalization was conducted. Pixels inside of the lung were normalized by subtracting their values with the mean intensities outside of the lung regions. To compare the correlation of the extent and density scores for four regions, the normalized pixels were averaged in each region and then we evaluated the correlation.

### 2.4. Correlation with RALE Score

Extent (0 to 4) and density (0 to 3) scores of pulmonary opacities were manually annotated by an experienced radiologist according to the guideline [12]. For the comparison of correlation between the extent and density scores and the mean intensity, we used a subset of testing set in Table 2 with a RALE score larger than 0 and the Pearson correlation [24] was calculated.

### 2.5. Data Description

The institutional review board (IRB) for human investigations at Massachusetts General Hospital (MGH, Boston, MA, USA), approved the study protocol with the removal of all patient identifiers from the images, and waived the requirement for informed consent, following the retrospective design of this study.

*(1) Segmentation.* Since anatomic segmentation of lungs is independent of radiographic abnormalities, we used two public datasets for training the segmentation model: RSNA pneumonia detection challenge dataset [25] and Japanese society of radiological technology (JSRT) dataset [26]. RSNA pneumonia detection challenge dataset consists of 568 CXRs from the tuberculosis chest images and the JSRT dataset consists of 257 CXRs. 

For evaluation of the segmentation model performance, we used 200 CXRs of 51 patients with COVID-19 pneumonia (Testing set in Table 2) obtained from three hospitals in South Korea including Kyungpook National University Hospital, Daegu Catholic University Hospital, and Yeungnam University Hospital.

*(2) Detection.* The carina and left hilum detection methods were trained on 704 CXRs from 166 COVID-19 patients (see training and validation sets in Table 2) between February and May 2020 from the same hospitals in South Korea, including Kyungpook National University Hospital, Daegu Catholic University Hospital, and Yeungnam University Hospital. The positions of the carina and left hilum, as landmarks to separate upper and lower lungs, were annotated under the supervision of a subspecialty chest radiologist with 13 years of clinical experience in thoracic imaging. For each CXR, a bounding box was placed around the left hilum. The inferior margin of the carina was also annotated with a point marker. A bounding box centered at the carina point was used for the training of the carina detection model.

*(3) Correlation*. To evaluate the clinical relevance of the proposed four-region segmentation method, CXRs were evaluated by the correlation of their RALE scores annotated by physicians. The physicians manually divided four-regions and the RALE score was measured by extent (0–4) and density (0–3) scores of pulmonary opacities in each region of the lung [12]. For each region, the correlation between the mean intensity and the corresponding extent and density scores of pulmonary opacities was computed.

### 2.6. Experimental Environment

Experimental environments were on Ubuntu 16.04 with a Tesla V-100 GPU, CUDA 9.0/cuDNN 7.0 (NVIDIA Corporation), and Keras 2.0 deep learning platform.

## 3. Results

Model performance for segmentation are listed in Table 3. The first to the fifth segmentation models were merged to the ensemble model. Model performance of the ensemble model including all models had the highest dice coefficient (0.908 ± 0.057) with significant statistical differences from Model 1 to 5 (All *p* < 0.05).

Figure 3 shows an example of advantages of the ensemble method for different quality of CXRs. The first to the last row in each column shows an input CXR, the ground truth mask, the ensemble result, and the five results predicted by the individual segmentation models. Figure 3a-1 shows a high quality CXR without medical device, substantial patient rotation, and over- or under- radiographic exposure. The five individual models gave consistent segmentations shown in Figure 3a-4–a-8). 

The CXR in Figure 3b-1 was challenging due to consolidation and/or atelectasis in the left lower lobe which obscures delineation of left lung hilum. Left lung hilum can also be obscured by dense perihilar opacities or marked patient rotation. Compared to the consistent results predicted by the first to third models as shown in Figure 3b-4–b-6 (0.929, 0.943, 0.934), the masks resulting from model 4 and 5 underestimated the area of right lung (0.817, 0.831). The ensemble could still reach a robust mask (0.933) as shown in Figure 3b-3. 

Figure 3c-1 shows a left chest tube traveling up to and obscuring visualization and detection of left hilum. Compared to the consistent results predicted by the third to fifth models as shown in Figure 3c-6–c-8) (0.883, 0.879, 0.903), the first and second models labeled areas outside of lung regions as shown in Figure 3c-4,c-5 (0.783, 0.885) due to extending into the right chest wall subcutaneous emphysema which has intensity similar to the right lung. The ensemble results gave a relative robust mask (0.899) as shown in Figure 3c-3.

Model performance for detection of left hilum and carina in terms of mean of average precision (mAP) was observed at 0.694. Figure 4 shows different examples for selection of a reference point to divide upper and lower lungs. Figure 4a shows an example with high confidence in detection result (left hilum: 0.94), where the center of the left hilum bound box is directly used as the reference horizontal level for the upper and lower lung region separation as shown in Figure 4b. In Figure 4c, the confidence of the detection result was low (left hilum: 0.56), and then the vertically 2 cm lower position of the carina bound box was used for the upper and lower region separation as shown in Figure 4d.

After normalization, the mean intensity of each region was correlated with the corresponding extent (0–4) and opacity (0–3) scores. Figure 5a–d shows the correlation of the extent score with mean intensities for each region, i.e., RUR, LUR, RLR, and LLR. For each region, the mean intensity increased as the extent scores increased. The correlation with the extent score for the LUR showed a strong positive linear relationship at 0.716 (<0.001) as shown in Figure 5c, and correlation values with the extent score for LUR, RUR, and RLR showed moderate positive linear relationship at 0.625 (<0.001), 0.454 (<0.001), and 0.457 (<0.001), respectively, as shown in Figure 5a,b,d. 

In case of density scores, the tendency that each mean intensity increased as the density scores increased was observed as shown in Figure 5e–h. The correlation with the density scores for RUR, LUR, RLR, and LLR showed moderate positive linear relationship at 0.553 (<0.001), 0.469 (<0.001), 0.506 (<0.001), and 0.465 (<0.001), respectively.

Distribution of mean intensity for each region is shown in Figure 6. Sum of left lung region is higher than that of right lung region. The mean intensity of LLR where heart is not segmented in the segmentation algorithm is lower than that of other regions.

## 4. Discussion

In the detection model for CXRs of anterior-posterior (AP), the detection inference value of the carina (average 0.743) was higher than that of the left hilum (average 0.467) because the label of the exact location of the left hilum is difficult in AP-type CXRs due to a wider longitudinal extent, overlap from cardio-mediastinal structures, obscuration from adjacent pulmonary opacities, and overlayed lines and tubes while labeling the location of the carina is easier due to less noise. It showed that the model performance highly depends on the quality of labeled data even using the same number of the training set.

In the validation of segmentation models, two different public datasets (RSNA pneumonia detection challenge and JSRT datasets) and the anonymized in-house dataset were used. On the public datasets, the five individual models showed sufficient dice coefficients around 0.958–0.967 due to the high radiographic quality of the PA CXR public datasets with low disease burden; both factors make it easier for each model to segment lung regions. However, in the anonymized dataset most CXRs for COVID-19 patients were captured with AP CXRs in the training dataset, which is common for severe patients. The utility of the AP CXRs was challenging due to lower radiographic quality, lower lung volumes, patient rotation, and a larger number of chest tubes, lines, and medical devices. To overcome these issues, the ensemble method was selected with segmentation models trained with different conditions, which was very effective to remove false positive and negative regions. In the training of segmentation models, different augmentation properties and backbone networks with the ensemble lead to robust lung masks under various situations such as patient position, image quality, and intubated patients. To avoid data issues generated from different portable devices and patient sizes and postures that never been exposed while training, Gaussian noise and distortion (non-rigid) transform-based augmentations were applied for the performance improvement because the patient size varies the noise level and the deformation can mimic the various patient postures and devices. Multiple backbone networks and ensemble strategy were robust to intubated patients and low contrast CXRs.

Correlation of the extent and density scores of pulmonary opacities with the mean intensities for each lung region showed positive linear relationships as shown in Figure 3. For RLR, the correlation of extent score with a mean intensity showed a strong positive relationship in Pearson correlation of 0.716 (*p* < 0.005). Dense basilar opacities in COVID-19 pneumonia, likely related to severe airspace opacification (or consolidation on CT images), obscure the lung margins at their interface with hemidiaphragm and cardio-mediastinal structures (see the obscured lower lungs in Figure 4a), which can degrade the performance of segmentation as well as the evaluation of the extent and density of pulmonary opacities. Such opacities may require additional dedicated training datasets which were not available to our model. 

Although we showed the Pearson correlation of the segmented regions with the extent and density scores of pulmonary opacities, the proposed method still has a great potential combined with various clinical applications such as classifying COVID-19 pneumonia and regular pneumonia. For the development of clinical methods, the segmentation model will be a crucial pre-processing tool for extracting lung regions in CXRs. In the future, combined with the proposed method as a pre-clinical step, we will develop an automatic prediction method of the RALE and the severity prediction model for COVID-19 patients.

## 5. Conclusions

In this paper, we proposed a fully automated four-region lung segmentation method in CXRs for COVID-19 patients and validated the method by the sub-regional correlation with the RALE value for clinical use. To evaluate the feasibility of the proposed method as one of the pre-processing methods in CXR, we demonstrated the positive correlation between intensities of segmented regions and the extent and density scores of pulmonary opacities. The ensemble strategy using five models showed the high performance of dice coefficient compared to a single model. Future work will focus on the automatic prediction of the RALE combined with the proposed segmentation method, and perform clinical evaluations using CXRs from multiple sites caring for COVID-19 patients.

## Figures and Tables

**Figure 1 diagnostics-12-00101-f001:**
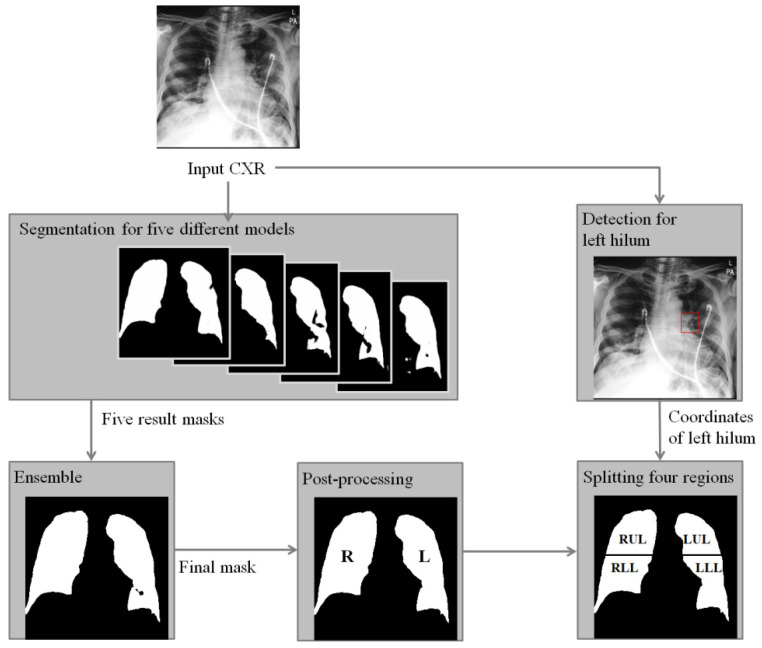
A flowchart of the proposed algorithm for segmentation of zones of the lung in CXR of COVID-19 patient. Right (R) and left (L) lung masks are generated by an ensemble method based on the majority voting from five lung masks predicted by models trained with different conditions. Then, left hilum and carina are detected and used to find a central point to split the whole lung into upper and lower regions. Finally, right upper lung (RUR), right lower lung (RLR), low upper lung (LUR), and left lower lung (LLR) are obtained.

**Figure 2 diagnostics-12-00101-f002:**
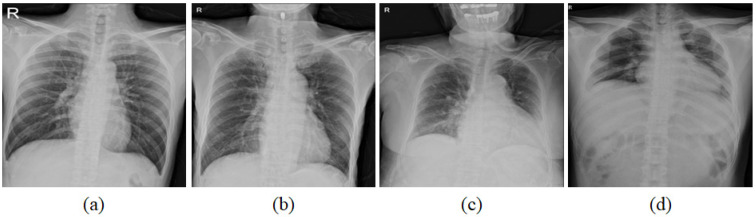
With the same density score as zero annotated by physician, mean intensities of lungs in CXRs are (**a**) 39.8, (**b**) 34.4, (**c**) 16.6, and (**d**) 13.2, respectively.

**Figure 3 diagnostics-12-00101-f003:**
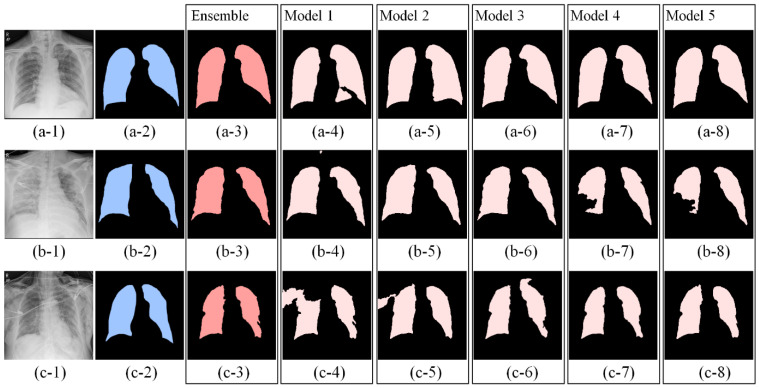
An example of advantages of the ensemble method for different quality of CXRs. The first to last row in each column shows an input CXR (**a**,**b**,**c-1**), a ground truth mask (**a**,**b**,**c-2**), an ensemble result (**a**,**b**,**c-3**), and five results predicted by the first to fifth model. (**a-1**) A clear CXR that shows none of severe noise from a portable device and obstacles like medical devices, (**b**) a lung mask of (**a-1**,**a-3**) an ensemble mask from the first to the fifth masks (**a-4**–**a-8**). Dice coefficients of (**a-3**–**a-8**) are 0.955, 0.928, 0912, 0.948, 0.948, and 0.948, respectively. (i) An CXR showing severe blurry within both lung regions due to lung opacity, (**b-2**) a lung mask of (**b-1**,**b-3**) an ensemble mask from the first to the fifth masks (**b-4**–**b-8**). Dice coefficients of (**b-3**–**b-8**) are 0.955, 0.928, 0912, 0.948, 0.948, and 0.948, respectively. (**c-1**) An CXR showing sever noise generated from a portable device, (**c-2**) a lung mask of (**c-1**,**c-3**) an ensemble mask from the first to the fifth masks (**c-4**–**c-8**). Dice coefficients of (**c-3**–**c-8**) are 0.899, 0.783, 0.885, 0.883, 0.879, and 0.903, respectively.

**Figure 4 diagnostics-12-00101-f004:**
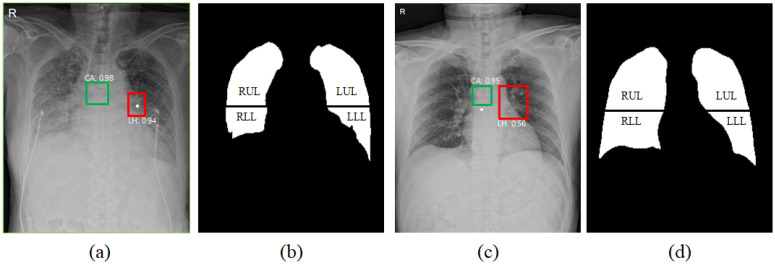
An example of detection results for the left hilum colored at red and carina colored at green and, dividing segmented lung mask into upper and lower lungs, i.e., RUR, LUR, RLR, and LLR with a reference point colored at while. (**a**) Detection results for the left hilum (confidence: 0.94) and the carina (0.98). (**b**) A center point of the detection box for the left hilum is used as the reference point to divide upper and lower lungs. (**c**) Detection results for the left hilum (0.56) and carina (0.95). (**d**) A location down to approximately 2 cm vertically from a center point of the detection box for the carina is used as the reference point to divide upper and lower lungs.

**Figure 5 diagnostics-12-00101-f005:**
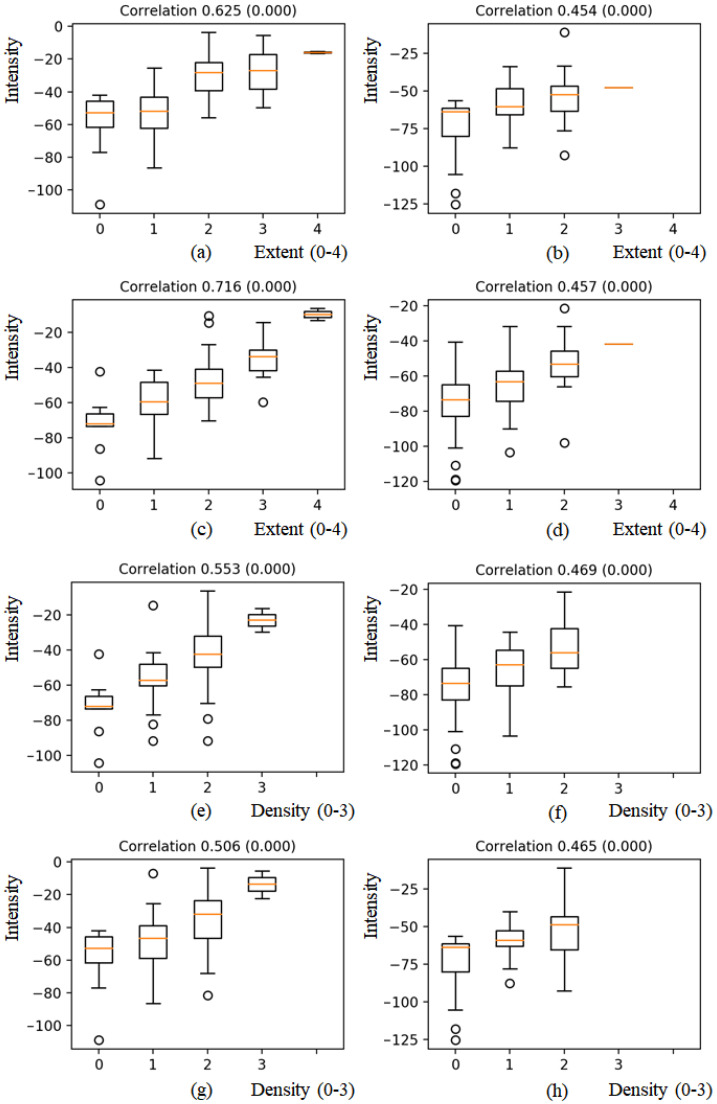
Boxplots of mean intensities with extent scores (0–4) and density scores (0–3) of pulmonary opacities for four-regions. (**a**) and (**e**) RUR, (**b**) and (**f**) LUR, (**c**) and (**g**) RLR, (**d**) and (**h**) LLR. For each region, the mean intensity increased as the extent and density scores increased.

**Figure 6 diagnostics-12-00101-f006:**
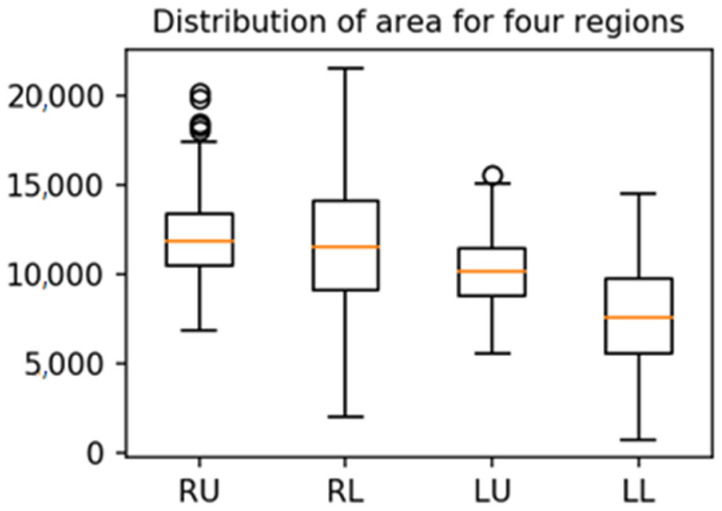
Boxplots of mean intensities for four-regions. The mean intensity of LLR, where the heart is not included in the segmentation algorithm, is lower than the intensities of other regions.

**Table 1 diagnostics-12-00101-t001:** Conditions for training different segmentation models.

Model	Backbone	Pre-Proc.	Augmentation
Model 1	Efficient0	N/A	DA
Model 2	Efficient0	HE	DA
Model 3	Efficient0	HE	DA + Gaussian noise (0.5) + gamma correction (0.5) + grid distortion (0.1) + elastic transform (0.1) + affine transform (0.1)
Model 4	Efficient7	HE	DA + Gaussian noise (0.5) + gamma correction (0.5)
Model 5	Efficient7	HE	DA + Gaussian noise (0.5) + gamma correction (0.5) + grid distortion (0.1) + elastic transform (0.1) + affine transform (0.1)

Abbreviations: HE, histogram equalization; DA, default augmentation (horizontal flip: 0.5, rotation: a range of ±25°, random contrast: 0.1, random brightness 0.1, gamma correction: 0.1, Gaussian noise: 0.1, contrast limited adaptive histogram equalization 0.1).

**Table 2 diagnostics-12-00101-t002:** Demographics of the dataset for carina and left hilum detection.

	Training Set (*n* = 551)	Validation Set (*n* = 153)	Testing Set (*n* = 200)
Patient	124	42	51
Age	68.3 ± 14.8	59.5 ± 16.2	54.3 ± 18.4
Male	53 (42.7%)	16 (38.0%)	23 (54.7%)
RALE	9.9 ± 10.7	3.9 ± 6.7	4.2 ± 6.2
Death	43 (34.6%)	2 (4.7%)	4 (9.5%)

**Table 3 diagnostics-12-00101-t003:** Performance comparison with single and ensemble model in terms of dice coefficient for the anonymized dataset in South Korea.).

No.	Model	Mean ± Std.
1	Model 1	0.874 ± 0.057 *
2	Model 2	0.854 ± 0.072 *
3	Model 3	0.873 ± 0.089 *
4	Model 4	0.888 ± 0.084 *
5	Model 5	0.889 ± 0.079 *
6	Ensemble	0.900 ± 0.074

(* Indicates a significant difference between an ensemble and other models, *p* < 0.05).

## Data Availability

https://github.com/younggon2/Research-Segmentation-Lung-CXR-COVID19 (accessed on 30 December 2021).

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
