# Peer review of "Deep Learning-Based Four-Region Lung Segmentation in Chest Radiography for COVID-19 Diagnosis"

_diagnostics, 2022, doi:10.3390/diagnostics12010101_

Round 1

Reviewer 1 Report

The article titled: Deep Learning-based Four-region Lung Segmentation in Chest Radiography for COVID-19 Diagnosis submitted to Diagnostics, presents very interesting, novel and relevant study. Authors proposed deep learning models to assist the diagnosis of lung severity using CXR. Procedure and protocol related with segmentation model is extensively described. Results present great potential including clinical future application. 

I recommend paper to publication in present form. 

Author Response

I recommend paper to publication in present form.

-> Re1. Thank you for your decision, sincerely.

Reviewer 2 Report

Dear Authors,

The manuscript overall quality is very good. I have a few concerns about this paper.

1, Could this methods distinguish COVID-19 pneumonia and regular pneumonia induced opacities?

2, What is the COVID-19 positive cases detecting accuracy like by using this method? The whole manuscript is trying to address COVID-19 severity diagnosis problem, but I think reader might be more interested in COVID-19 screening accuracy by using this new method.

Author Response

Reviewer #2: 

Dear Authors,

The manuscript overall quality is very good. I have a few concerns about this paper.

1, Could this methods distinguish COVID-19 pneumonia and regular pneumonia induced opacities?

> Re 2-1. Thanks for your comment. We think that classifying both classes (COVID-19 pneumonia and regular pneumonia) would be one of the important tasks for clinical uses in X-ray. By the way, the purpose of the proposed method is to segment the whole lung and split it four regions, which can be used as a pre-processing for further applications such classifying COVID-19 pneumonia and regular pneumonia.

To clarify this point, we modified our manuscript as follows in Result (11 of 12 pp.)

“Although we showed the Pearson correlation of the segmented regions with the extent and density scores of pulmonary opacities, the proposed method still has a great potential combined with various clinical applications such as classifying COVID-19 pneumonia and regular pneumonia.”

2, What is the COVID-19 positive cases detecting accuracy like by using this method? The whole manuscript is trying to address COVID-19 severity diagnosis problem, but I think reader might be more interested in COVID-19 screening accuracy by using this new method.

> Re 2-2. Thanks for your comment. I fully understand that readers are more interested in COVID-19 screening than others. In our study, we focused on how to segment the regions in X-ray more precisely. Thus, we did not thoroughly overview the COVID-19 positive cases. Calculating each correlation of severity with the four regions is just to validate how the proposed method segment the four regions clearly for X-ray of patients with COVID-19 that image quality is inferior than other quality of X-ray.

To clarify this point, we modified our manuscript as follows in Conclusion (11 of 12 pp.)

“To evaluate the feasibility of the proposed method as one of the pre-processing methods in CXR, we demonstrated the positive correlation between intensities of segmented regions and the extent and density scores of pulmonary opacities.”
